# Effects on *Pinus densiflora* Seedlings as Affected by Different Container Growth Conditions

**DOI:** 10.3390/ijerph17103565

**Published:** 2020-05-19

**Authors:** Jae Hwan Kim, Byoung-Youn Kang, Jungho Ryu, In-Hyun Nam

**Affiliations:** 1Department of Human Environment Design, College of Science, Cheongju University, Cheongju 28503, Korea; smileeye77@cju.ac.kr; 2Chungbuk Farm Company Farming Association Corporation, Okcheon 29061, Korea; cbfarm@naver.com; 3Geologic Environment Research Division, Korea Institute of Geoscience and Mineral Resources (KIGAM), Daejeon 34132, Korea; jryu@kigam.re.kr

**Keywords:** negative phototropism, seedling growth, root collar diameter, root status, *Pinus densiflora* seedling

## Abstract

The purpose of this study was to determine the effects that different container conditions have on *Pinus densiflora* seedling growth. Under greenhouse cultivation, there were no statistical differences observed in plant height and the number of branches; however, significant differences in root collar diameter and root status were observed. In control container growth conditions, the roots grew in an abnormal spiral shape, while in negative phototropism container growth conditions the roots grew in a vertical shape. In outdoor cultivation, seedlings in various container growth conditions showed significant differences. The seedlings that were grown in negative phototropism container growth conditions showed the greatest increases in height, number of branches, root collar diameter, and root growth. This study determined that seedling roots in negative phototropism container growth conditions grew vertically, thus displaying successful rooting when they were transplanted outdoors. This resulted in favorable measurements in height, number of branches, root collar diameter, and root growth.

## 1. Introduction

With increasing interest in the environment, the trend of using plants for landscaping is moving away from an emphasis on the quantity of limited tree species toward high quality, standardized plant production [1]. The systematic production of healthy, good quality, standardized plants for landscaping is a crucial process used to increase initial plant rooting promoting growth; container production methods are often used to achieve this [2]. The production of healthy seedlings is also critical to improve initial rooting and growth after transplantation, with soil mix and container type for seedling production emerging as important issues [3,4].

In general, soil improver treatment for landscape plant production is widely used as a way to promote root growth by improving the breathability and permeability of soil [5]. However, studies on methods used to promote seedling growth according to the growth conditions of container they are grown in are still insufficient. Technologies and research regarding container growth conditions are crucial because containers not only have a direct influence on the root shape formed in the container during the nursery period, but also have significant effects on short-term and long-term growth, even after planting [6,7].

Growing plants in containers has many advantages, including allowing planting with no root damage, similar to growing plants in bare ground, thus decreasing plant defects due to easier rooting after transplantation and decreased seasonal variation effects. Furthermore, plants with uniform dimensions can be produced in large numbers and specialized skills are not required for root harvesting [8,9]. Owing to these advantages of growing plants in containers, interest in containers is rising in domestic sites for landscape plants, requiring further development of containers using various shapes and materials [2]. However, the development of abnormal spiral roots during seedling production using general containers interferes with rooting after planting [10], which also requires solutions. When the growing status of the *Pinus densiflora* container seedlings afforested in the Goseong (Kangwon Province, Korea) forest fire site was examined, ground growth was adequate, but the roots of most *Pinus densiflora* seedlings were found to grow poorly, which is a problem that needs to be addressed [4,11]. Therefore, in the present study, we examined the growing characteristics of *Pinus densiflora* according to different container growth conditions.

Recommendations for the development of appropriate container growth conditions were previously reported, including plant production using containers of various shapes, physical and chemical cultivation techniques, differences in growth by container type, rooting after transplantation according to the root system development, and ecological differences of fine root development [7,12,13]. However, this study aimed to address and investigate the effects of container growth conditions (CGCs) on *Pinus densiflora* seedlings and their growth characteristics.

## 2. Materials and Methods

### 2.1. Materials

#### 2.1.1. Experimental Species

We selected *Pinus densiflora* as the experimental species (*Pinus densiflora* Siebold and Zucc.) among evergreen trees, which is widely used for landscaping and afforestation in forest fire sites (Goseong, Kangwon Province, Korea) [7], and allows growth result determination in a short period due to accelerated growth and root system development. For this experiment, we selected two-year-old plug *Pinus densiflora* seedlings with specific heights, numbers of branches, root collar diameters, and root lengths. The growth status of the seedlings used in this experiment is shown in Table 1.

#### 2.1.2. Soil Conditions

The culture soil used in this study was bed soil for gardening (cocopeat 68.0%, peatmoss 15.0%, perlite 7.0%, vermiculite 6.0%, and zeolite 4.0%) and the soil of the ground experiment groups was sandy clay loam soil that did not contain many nutrients and was composed of 94.7% sand, 4.4% silt, and 0.9% clay. The analysis results of the physical and chemical properties of the soil are shown in Table 2.

### 2.2. Methods

#### 2.2.1. Container Growth Conditions (CGCs)

The containers used in this study were divided into control and negative phototropism containers and further subdivided into six categories in order to examine the effects that the existence or absence of a root-turning bump inside the container and bump thickness had on the growth of *Pinus densiflora* (Figure 1). We prepared the control container growth condition (CGC-1) with a top diameter of 18 cm, a bottom diameter of 16 cm, a depth of 20 cm, and a capacity of 4 L. For container growth condition 2 (CGC-2), a 5 mm thick plastic root-turning bump was installed inside CGC-1. For container growth condition 3 (CGC-3), a 10 mm thick plastic root-turning bump was installed inside CGC-1. For container growth condition 4 (CGC-4), a vertical striped slit was made on the surface of CGC-1 so that the roots would receive sunlight. For container growth condition 5 (CGC-5), a 5 mm thick plastic root-turning bump was installed inside CGC-4. For container growth condition 6 (CGC-6), a 10 mm thick plastic root-turning bump was installed inside CGC-4. Four root-turning bumps were installed uniformly in the vertical direction in the negative phototropism container growth conditions (Figure 1).

#### 2.2.2. Planting, Transplantation, and Management of the Container Growth Conditions

Each 4 L container was filled with the same bed soil, in which the prepared *Pinus densiflora* seedlings (Table 1) were planted. The different growth conditions according to container type were arranged in a greenhouse using the three-repetition random placement method. A total of 90 stocks, consisting of 3 groups of 5 stocks for each of the 6 treatments, were planted on 25 April 2017 and grown for 12 months to 20 April 2018. They were supplied with 3 L of water once every 3 days using an automatic sprinkler irrigation system so that the soil surface would not become dry; topdressing was not performed.

All the *Pinus densiflora* seedlings from all the different CGCs grown in the greenhouse were transplanted to farm ground in Iwon-ri, Iwon-myeon, Okcheon-gun, Chungcheongbuk-do, South Korea on 21 April 2018. The 2018 annual average temperature in this area was 11.6 °C, and the annual precipitation was 1409.5 mm [14]. The ground type was a tree nursery on general flatland (12 m × 8 m) that was well-drained and received natural light due to the absence of surrounding objects. Two of the five stocks for each CGC used in the greenhouse experiment were used for root measurement on 13 November 2017. For the in-ground experiment, 3 stocks were planted using the three-repetition random placement method according to the CGC for a total of 54 weeks, excluding the seedlings used for root measurement. After ground transplantation, all containers were sufficiently irrigated until water flowed down to the surface; no separate irrigation and topdressing were thereafter performed.

### 2.3. Measurement and Analysis

To examine the growth characteristics of the *Pinus densiflora* seedlings according to their container growth conditions, we measured the moisture content, height, number of branches, and root collar diameter seven times between 13 May 2017 and 27 October 2018 every 3 months. For root growth, the root length and dry weight were examined twice between 13 November 2017 and 27 October 2018 every 6 months. First, the soil moisture content was determined by measuring the transit time of an electromagnetic pulse launched along a parallel metallic probe of Time Domain Reflectometry (TDR) buried in the soil. For height, the vertical length from the surface of the soil to the top of the plant was measured using a height measuring instrument (FA11, Kyelim Landscape Coporation, Daegu, Korea). For number of branches, the total numbers of generated branches were counted. These heights and numbers of branches were compared with those measured at the time of initial growth. The root collar diameter was regularly measured using Vernier calipers (530–100) from Mitutoyo Co., Ltd. according to marks on the stem that came in contact with the ground surface. Finally, the roots were sufficiently washed with running water so that all dirt on the surface was removed and then dried completely in a thermostat (CT-DO 72) at 65 °C. The lengths and dry weights were then measured. The measurement results were analyzed using the statistics package application IBM SPSS Statistics Version 25 (IBM Corporation, Armonk, NY), and the average intervals were analyzed using Duncan’s multiple range test.

## 3. Results and Discussion

### 3.1. Physical and Chemical Properties of Soil

#### Moisture Content

The moisture content of soil for each CGC was measured, with the results presented in Table 3. The three measurements taken in May, July, and November 2017 showed similar levels, with no significant differences between any of the CGCs. When they were measured on 12 September 2017, CGC-1, CGC-2, and CGC-3 showed higher moisture contents than CGC-4, CGC-5, and CGC-6, with the differences being statistically significant (Table 3). The negative phototropism container growth conditions (CGC-4, CGC-5, and CGC-6) showed somewhat low moisture contents because they had slits on the container surfaces, but their moisture contents were not low enough to inhibit growth. Three measurements taken between 21 June 2018 and 27 October 2018 showed similar moisture content levels among all CGCs due to the effect of rainfall [14]; no factor was observed to influence the plant growth.

### 3.2. Growth Characteristics of the Shoots of Pinus densiflora Seedlings

#### 3.2.1. Height and Branch Growth

The heights and numbers of branches of *Pinus densiflora* seedlings according to the CGCs they were grown in were measured, with the results shown in Table 4 and Figure 2. They were measured in the greenhouse four times between 13 May 2017 and 13 November 2017, showing similar levels with no statistically significant differences among the CGCs. However, CGC-1, CGC-2, and CGC-3 showed higher trends than the negative phototropism containers (CGC-4, CGC-5, and CGC-6), presumably due to the low moisture contents (Table 3) of CGC-4, CGC-5, and CGC-6 due to their slits on the container surfaces. In the greenhouse experiment, small differences in the size of root-turning bumps installed inside the containers were observed; the existence or absence of the surface slit appeared to slightly affect growth in terms of height and number of branches, but there were no statistically significant differences observed. A previous study reported that bumps designed inside containers force roots to grow downward and prevent roots from spreading sideways [15]; however, different results were obtained in the present study.

The results of ground measurements taken three times between 21 June 2018 and 27 October 2018 supported that the heights and numbers of branches were higher in the negative phototropism CGCs (CGC-4, CGC-5, and CGC-6) than in the control CGCs (CGC-1, CGC-2, and CGC-3). The roots of the control container seedlings formed spiral shapes inside the container, resulting in a negative effect on height and branch growth [6]. However, the roots of the negative phototropism CGCs grew vertically, with this rooting maintained after transplantation, resulting in a positive effect on height and branch growth. The last measurement, taken in October 2018, showed that height and branch growth were 22.4 cm and 13.3 for the CGC-1, 22.8 cm and 17.3 for CGC-2, 23.0 cm and 18.5 for CGC-3, 39.7 cm and 22.3 for CGC-4, 38.8 cm and 21.6 for CGC-5, and 37.0 cm and 20.7 for CGC-6 (Table 4 and Figure 2). The differences between CGC-4, which showed the highest values, and CGC-1, which showed the lowest values, were 17.3 cm in height and 9.0 in the number of branches. These findings were consistent with previous results [16] reporting that seedlings displayed a good balance of growth when their roots were in a container with a slit, inducing partial air pruning, vertical growth, and subsequent natural development of rootlets, with ground growth being promoted even after the seedlings were transplanted. The results showed that the negative phototropism CGCs (CGC-4, CGC-5, and CGC-6) were effective in regard to height and branch growth, and the root-turning bumps inside the container did not exhibit greater growth effects than the slit. Therefore, in terms of landscape plant production, negative phototropism CGCs are considered to be effective.

#### 3.2.2. Root Collar Diameter

The diameters of growth at the root collar of *Pinus densiflora* according to the CGCs they were grown in were measured, with the results shown in Figure 3. The results of seven measurements taken from May 2017 to October 2018 showed a clear difference between the two different types of container (control versus negative phototropism). The negative phototropism CGCs (CGC-4, CGC-5, and CGC-6) showed more distinct root collar diameter growth than the control CGCs (CGC-1, CGC-2, and CGC-3) (Figure 3), possibly because as the roots of the seedlings grown in containers that received sunlight came out through the slits on the containers, air pruning exerted a greater effect on the growth of the root collar diameter than the height [6].

Four measurements taken in the greenhouse from 13 May 2017 to 13 November 2017 showed statistically significant differences among the CGCs. The root collar diameters of the negative phototropism CGCs (CGC-4, CGC-5, and CGC-6) were greater than those of the control CGCs (CGC-1, CGC-2, and CGC-3). The control CGCs did not show significant differences, but the CGCs with root-turning bumps showed an increasing trend (CGC-3 ≤ CGC-4), whereas the negative phototropism CGCs were not greatly affected by the existence or absence of root-turning bumps. A previous study reported that bumps designed inside containers force roots to grow downward and prevent roots from spreading sideways [15,17]. In the present study, similar results were obtained in the CGCs, but the bumps showed no effect in the phototropism CGCs.

Three measurements taken between 21 June 2018 and 27 October 2018 after ground-planting showed statistically significant differences among the CGCs. As with the greenhouse experiment results, the negative phototropism CGCs showed greater root collar diameter growth. From the last measurement taken in October 2018, the ascending order of root collar diameter was as follows: CGC-1 (11.0 mm) ≤ CGC-2 (11.7 mm) ≤ CGC-3 (12.3 mm) < CGC-6 (17.5 mm) ≤ CGC-5 (17.7 mm) ≤ CGC-4 (18.0 mm). The root collar diameter of CGC-4 was 7.0 mm greater than that of CGC-1, suggesting that the negative phototropism container was more effective in regard to the growth of root collar diameter than the control container, and the surface slits exhibited greater growth effects than the root-turning bumps inside the containers. Therefore, the use of negative phototropism CGCs are expected to produce more homogeneous, healthier trees.

### 3.3. Growth Characteristics of the Lower Part of Pinus Densiflora Seedlings

#### Root Growth

The root growth of *Pinus densiflora* seedlings according to the growth conditions of the containers they were grown in was measured, with the results shown in Figure 4. The measurements of root length taken on 13 November 2017 showed statistically significant differences among the CGCs. CGC-1 showed the longest roots and CGC-6 showed the shortest roots. The roots were shorter in the negative phototropism CGCs, possibly because the roots of the CGC-1 were longer than the container depth of 2.0 cm because the roots grew in a spiral shape, whereas the roots in the negative phototropism container growth conditions (CGC-4) grew vertically and could only reach a maximum length of 20 cm due to the length of the container (Figure 5). Among the CGCs, the roots of CGC-2 and CGC-3 with root-turning bumps were shorter than those of CGC-1, but the effect of the bumps was not significant (Appendix A). The differences between the 5 mm and 10 mm root-turning bumps were also insignificant, indicating that their role in preventing the spiral growth of roots is inadequate (S-1). The dry weight results showed statistically significant differences among the CGCs; these results were opposite to the root length results. The dry weights of the negative phototropism CGCs were greater than those of the CGC-1 with long root lengths, because the roots were pruned by air and withered in the negative phototropism containers, thereby preventing root length growth and the promotion of root growth via photosynthesis [16]. These results suggest that the negative phototropism CGCs has a positive effect on root growth.

The measurements taken on 27 October 2018 after the seedlings were planted in the ground showed statistically significant differences in root length among the container growth conditions. The negative phototropism CGCs (CGC-6 ≤ CGC-5 ≤ CGC-4) showed longer roots than the other CGCs (CGC-1 ≤ CGC-2 ≤ CGC-3), which was opposite to the results obtained from greenhouse measurements (Figure 4a). This was because there was low rooting after transplantation in the control CGCs due to the spiral growth of the roots, whereas in the negative phototropism CGCs, vertically grown roots exerted a positive effect on rooting. The dry mass of roots showed statistically significant differences among the CGCs, with the ascending order of dry mass as follows: CGC-1 ≤ CGC-2 ≤ CGC-3 < CGC-6 ≤ CGC-5 ≤ CGC-4 (Figure 4b). Thus, the negative phototropism CGCs showed better root growth, with the root growth of CGC-4 being particularly effective. Marler and Willis (1996) reported that nonideal roots, such as spiral roots, exerted negative effects on rooting and growth after planting seedlings, which was consistent with the results of the present study. Therefore, the healthy development of roots would be a good indicator for predicting seedling quality and determining survival and growth rate after planting at the site [18]. The results suggested that growing seedlings in negative phototropism CGCs is an effective method, with vertical root growth and physiologically healthy plant growth after planting.

## 4. Conclusions

The growth characteristics and effects of the six different container growth conditions (CGCs) of *Pinus densiflora* seedlings were examined. The negative phototropism CGCs demonstrated greater effectiveness in regard to the growth of *Pinus densiflora* seedlings compared to the control CGCs, showing positive results in height, number of branches, root collar diameter, and root growth. Therefore, the use of negative phototropism CGCs is expected to produce more healthy trees. The present findings indicated that negative phototropism CGCs are promising for use as an efficient, bioremediation-based method of forest fire site restoration. This study was limited to evergreen *Pinus densiflora*; different results may be obtained with other species. Therefore, further research on various additional species and CGCs should be conducted.

## Figures and Tables

**Figure 1 ijerph-17-03565-f001:**
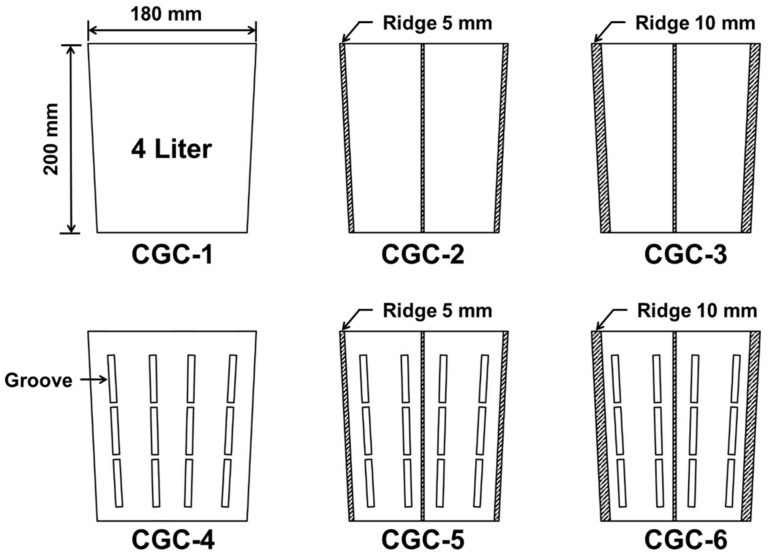
Six different container growth conditions (CGCs) used in this study. CGC-1: Control container without modification; CGC-2: CGC-1 with inside 5 mm thick root-turning bump; CGC-3: CGC-1 with inside 10 mm thick root-turning bump; CGC-4: negative phototropism container with a vertical striped slit was made on the surface of CGC-1; CGC-5: CGC-4 with inside 5 mm thick root-turning bump; CGC-6: CGC-5 with inside 10 mm thick root-turning bump.

**Figure 2 ijerph-17-03565-f002:**
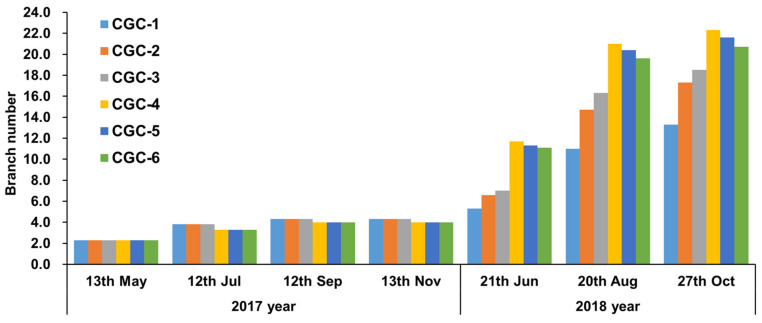
*Pinus densiflora* seedling branch numbers affected by different container growth conditions (CGCs). CGC-1: Control container without modification; CGC-2: CGC-1 with inside 5 mm thick root-turning bump; CGC-3: CGC-1 with inside 10 mm thick root-turning bump; CGC-4: negative phototropism container with a vertical striped slit made on the surface of CGC-1; CGC-5: CGC-4 with inside 5 mm thick root-turning bump; CGC-6: CGC-5 with inside 10 mm thick root-turning bump.

**Figure 3 ijerph-17-03565-f003:**
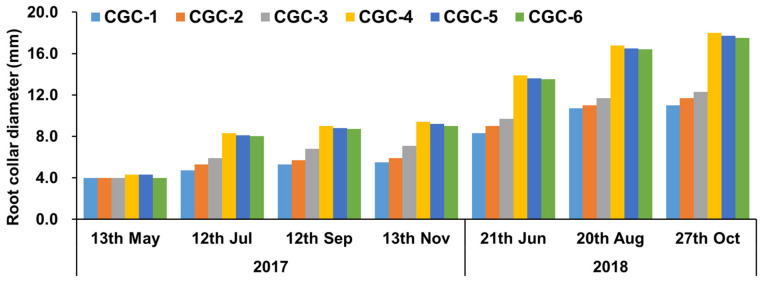
*Pinus densiflora* seedling root collar diameters (mm) affected by different container growth conditions (CGCs). CGC-1: Control container without modification; CGC-2: CGC-1 with inside 5 mm thick root-turning bump; CGC-3: CGC-1 with inside 10 mm thick root-turning bump; CGC-4: negative phototropism container with a vertical striped slit made on the surface of CGC-1; CGC-5: CGC-4 with inside 5 mm thick root-turning bump; CGC-6;: CGC-5 with inside 10 mm thick root-turning bump.

**Figure 4 ijerph-17-03565-f004:**
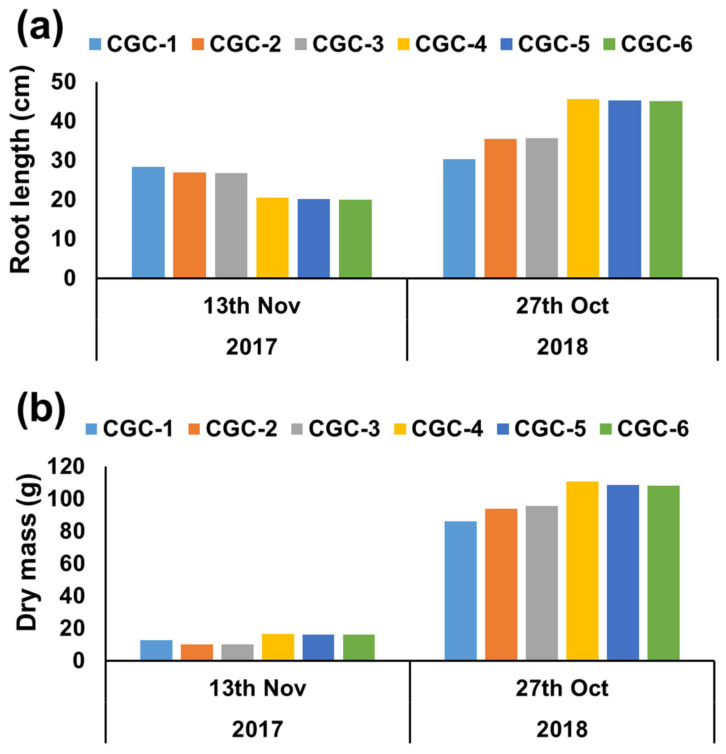
*Pinus densiflora* seedling root length (**a**) and root dry mass (**b**) affected by different container growth conditions. CGC-1: Control container without modification; CGC-2: CGC-1 with inside 5 mm thick root-turning bump; CGC-3: CGC-1 with inside 10 mm thick root-turning bump; CGC-4: negative phototropism container with a vertical striped slit made on the surface of CGC-1; CGC-5: CGC-4 with inside 5 mm thick root-turning bump; CGC-6: CGC-5 with inside 10 mm thick root-turning bump.

**Figure 5 ijerph-17-03565-f005:**
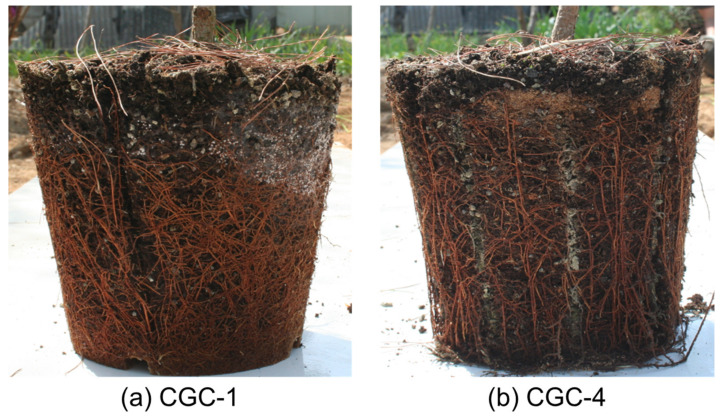
*Pinus densiflora* seedling root status affected by CGCs. In CGC-1 (**a**), the roots grew in a spiral shape, whereas the roots in CGC-4 (**b**) grew vertically.

**Table 1 ijerph-17-03565-t001:** Status of seedlings used in the experiment.

Flora	Scientific Name	Plant Status *			
		H (cm)	D (mm)	B (ea)	R (cm)
Evergreen	*Pinus densiflora* (Siebold and Zucc.)	10∼13	2∼3	4∼5	7∼7.5

Note: * Plant status: H, height; D, root collar diameter; B, branch number; R, root length.

**Table 2 ijerph-17-03565-t002:** Soil properties of the seedlings used for experiments.

Soil Property	pH	EC (dS/m)	P_2_O_5_ (mg/kg)	N (%)	O.M.* (%)	C.E.C.** (cmol/kg)	Exchangeable Cations (me/100 g)			
							Ca^2+^	Mg^2+^	K^+^	Na^+^
Container soil	5.9	0.9	364.0	0.5	32.6	21.8	7.4	4.7	4.4	3.8
Outdoor cultivation soil	6.6	0.1	36.4	0.1	0.3	4.1	2.2	1.3	0.3	0.2

Note: * O.M., organic matter; ** C.E.C., cation exchange capacity.

**Table 3 ijerph-17-03565-t003:** Moisture contents (%) affected by container growth conditions.

Container Growth Condition (CGC) *	2017				2018		
	13th May	12th July	12th September	13th November	21th June	20th August	27th October
CGC-1	12.3a **	21.7a	7.1a	15.4a	20.5a	10.0a	11.6a
CGC-2	12.5a	21.5a	6.9a	15.3a	20.8a	10.1a	11.6a
CGC-3	12.1a	21.2a	6.7a	15.8a	19.9a	9.9a	11.8a
CGC-4	11.5a	20.9a	4.2b	13.8a	19.3a	10.2a	11.5a
CGC-5	11.4a	20.6a	4.1b	13.6a	20.1a	9.9a	11.9a
CGC-6	11.2a	20.1a	4.0b	13.3a	20.3a	10.1a	11.7a

Note: * CGC-1: Control container without modification; CGC-2: CGC-1 with inside 5 mm thick root-turning bump; CGC-3: CGC-1 with inside 10 mm thick root-turning bump; CGC-4: negative phototropism container with a vertical striped slit made on the surface of CGC-1; CGC-5: CGC-4 with inside 5 mm thick root-turning bump; CGC-6: CGC-5 with inside 10 mm thick root-turning bump. ** Mean values with the same letter within columns are not significantly different at *p* = 0.05 according to Duncan’s test.

**Table 4 ijerph-17-03565-t004:** Height (cm) affected by container growth conditions of *Pinus densiflora* seedlings.

Container Growth Condition (CGC) *	2017				2018		
	13 May	12 July	12 September	13 November	21 June	20 August	27 October
CGC-1	13.0 **	17.3	19.3	19.9	21.0b	22.0b	22.4b
CGC-2	13.3	17.5	19.6	20.4	22.1b	22.3b	22.8b
CGC-3	13.3	17.6	19.7	20.6	22.5b	22.3b	23.0b
CGC-4	13.0	16.7	18.8	19.5	29.3a	36.7a	39.7a
CGC-5	13.0	16.6	18.6	19.3	28.5a	35.6a	38.8a
CGC-6	12.9	16.4	18.6	19.1	26.7a	33.3a	37.0a

Note: * CGC-1: Control container without modification; CGC-2: CGC-1 with inside 5 mm thick root-turning bump; CGC-3: CGC-1 with inside 10 mm thick root-turning bump; CGC-4: negative phototropism container with a vertical striped slit made on the surface of CGC-1; CGC-5: CGC-4 with inside 5 mm thick root-turning bump; CGC-6: CGC-5 with inside 10 mm thick root-turning bump. ** Mean values with the same letter within columns are not significantly different at *p* = 0.05 according to Duncan’s test.

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
