# Peer review of "Effects on Pinus densiflora Seedlings as Affected by Different Container Growth Conditions"

_ijerph, 2020, doi:10.3390/ijerph17103565_

Round 1

Reviewer 1 Report

The authors described here the effects that different types of containers have on Pinus densiflora seedlings growth; in particular it has been demonstrated that seedlings grown in specially manufactured negative phototropism containers showed the greatest increase in height, number of branches, root collar diameter and root growth.

The production of healthy seedlings is critical in order to improve the initial rooting and growth after transplantation and production of plants in containers seems to have many advantages. 

The description of the containers should be rewritten because is not well described and confused. What do you mean for "natural container"? Which is the material of the container? Figure 1 resolution is poor, it is not easy to read it. 

In 2.2.2 the sentence "They were irrigated using a sufficient automatic irrigation system so that the soil surface would not become dry" is not sufficiently reported. How do you irrigate the plants in the containers? 

Moisture content description lacks in details. Moreover there are some errors in the text. Please check them. 

I suggest to the authors to improve the quality of the images. 

Author Response

We appreciate the reviewer’s positive and encouraging feedback. We have modified the manuscript to address the reviewer’s criticisms, and hope that these changes are sufficient and acceptable. Please find the attached file. 

Reviewer 2 Report

 the effects that different types of containers on Pinus densiflora seedling growth were studied. Under greenhouse cultivation, there were no statistical differences observed in plant height and number of branches. This study determined that seedling roots in negative phototropism containers grew vertically, thus displaying
successful rooting when they were transplanted outdoor. This resulted in favorable measurements in height, number of branches, root collar diameter, and root growth.

comments this is a good manuscript in agricultural sciences

Author Response

We appreciate the reviewer’s positive feedback! We have modified the manuscript to address the reviewer’s criticisms, and hope that these changes are sufficient and acceptable.

Reviewer 3 Report

Referee report attached in PDF format

Author Response

(The authors gave the same response as above.)

Round 2

Reviewer 3 Report

See PDF attached
